# DIETing: Self-Supervised Learning with Instance Discrimination Learns Identifiable Features

**Attila Juhos**[*1], **Alice Bizeul**[*2], **Patrik Reizinger**[*1],
David Klindt[5], Randall Balestriero[4], Mark Ibrahim[6], Julia E. Vogt[2], Wieland Brendel[1]
`{patrik.reizinger, attila.juhos, wieland.brendel}@tuebingen.mpg.de`
`{alice.bizeul, julia.vogt}@inf.ethz.ch, klindt@cshl.edu,`
`rbalestr@brown.edu, marksibrahim@meta.com`

## Abstract

Self-Supervised Learning (SSL) methods often consist of elaborate pipelines with hand-crafted data augmentations and computational tricks. However, it is unclear what is the provably minimal set of building blocks that ensures good downstream performance. The recently proposed instance discrimination method, coined DIET, stripped down the SSL pipeline and demonstrated how a simple SSL algorithm can work by predicting the sample index. Our work proves that DIET recovers cluster-based latent representations, while successfully identifying the correct cluster centroids in its classification head. We demonstrate the identifiability of DIET on synthetic data adhering to and violating our assumptions, revealing that the recovery of the cluster centroids is even more robust than the feature recovery.

## 1 Introduction

Self-Supervised Learning (SSL) methods use unlabeled datasets to learn representations by solving an auxiliary task, thus bypassing time-consuming labelling efforts. Importantly, co-occurance–based SSL relies on positive data pairs (similar samples, e.g., an original sample and a transformed/augmented one) and negative data pairs (dissimilar samples, often randomly drawn from the dataset). Contrastive and non-contrastive learning, the two prominent families of SSL methods, utilize positives and negatives differently, though they are theoretically connected [Balestriero and LeCun, 2022]. Contrastive Learning (CL) [Chen et al., 2020, Zimmermann et al., 2021, von Kügelgen et al., 2021, Lyu et al., 2021, Eastwood et al., 2023] attracts positive pairs' and repels negative pairs' representations. Non-contrastive learning [Bardes et al., 2021, Zbontar et al., 2021, Mialon et al., 2022] only uses positive pairs, and avoids representation collapse with strategies such as momentum encoders or covariance regularization. Unfortunately, the many actively developed Self-Supervised Learning methods with such computational tricks potentially hinder selecting the best performing and simplest SSL method for a given task. Recently, Ibrahim et al. [2024] proposed DIET, a SSL method that strips away unnecessary details by reducing the auxiliary task to a simple instance classification paradigm, and showed competitive performance on small datasets.

Identifiability theory, particularly Independent Component Analysis (ICA) [Comon, 1994, Hyvarinen et al., 2001] studies guarantees of probabilistic models to recover the ground-truth latent variables in a probabilistic latent variable model (LVM). Recent advances in nonlinear ICA theory proposed multiple self-supervised/weakly supervised models with identifiability guarantees [Hyvarinen et al., 2019, Gresele et al., 2019, Khemakhem et al., 2020a, Hälvä et al., 2021, Hyvarinen and Morioka, 2016, Khemakhem et al., 2020b, Locatello et al., 2020, Morioka and Hyvarinen, 2023, Morioka et al.,

---

[*]Joint first authorship; [1]Max Planck Institute for Intelligent Systems, Tübingen AI Center, ELLIS Institute, Tübingen, Germany; [2]Department of Computer Science, ETH Zürich and ETH AI Center, ETH Zürich, Zürich, Switzerland; [4]Department of Computer Science, Brown University, Rhode Island, USA; [5]Cold Spring Harbor Laboratory, Cold Spring Harbor, New York, USA; [6]FAIR, META, New York, USA;

38th Conference on Neural Information Processing Systems (NeurIPS 2024).

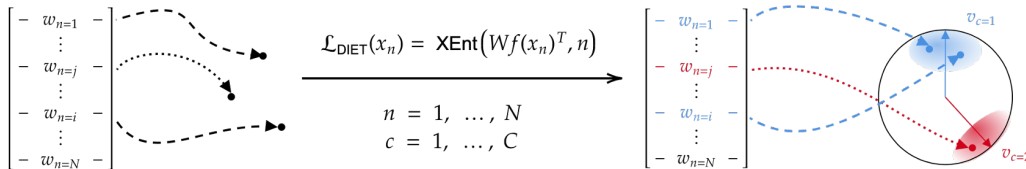

Figure 1: **DIET [Ibrahim et al., 2024] learns identifiable features**: DIET learns a linear $(N \times d)-$dimensional classification head $\boldsymbol{W}$ on top of a nonlinear encoder $\boldsymbol{f}$ through an instance discrimination objective (1). For unit-normalized $\boldsymbol{f}(\boldsymbol{x}_n)$, DIET maps samples and their augmentations close to the cluster vector $\boldsymbol{v}_c$ corresponding to the class as if sampled from a von Mises-Fisher (vMF) distribution, centered around the cluster vector. In case of duplicate samples, i.e., matching class labels, the corresponding rows of $\boldsymbol{W}$ will be the same, as shown for $\boldsymbol{x}_1$ and $\boldsymbol{x}_i$ with $\boldsymbol{w}_1 = \boldsymbol{w}_i$

2021]. Several papers study a contrastive scenario, [Hyvarinen and Morioka, 2016, Hyvarinen et al., 2019, Zimmermann et al., 2021, von Kügelgen et al., 2021, Rusak et al., 2024], providing a possible theoretical explanation for CL's practical success.

Our paper investigates whether DIET's competitive performance can be explained by identifiability theory. We model the data generating process (DGP) in a new, cluster-based way, and show that DIET's learned representation is linearly related to the ground truth representation. We also show how DIET's classification head recovers the cluster centroids, a connection to clustering that is absent from prior identifiability works for Self-Supervised Learning. Unlike other SSL solutions such as SimCLR [Chen et al., 2020], BYOL [Grill et al., 2020], BarlowTwins [Zbontar et al., 2021], or VICReg [Bardes et al., 2021], DIET's training objective applies to the same representation that is used post-training for solving downstream tasks. More precisely, no projector network is removed post-training. This implies that our theoretical guarantees directly apply to the SSL representation being used post-training, as opposed to other identifiability results in SSL [Zimmermann et al., 2021, von Kügelgen et al., 2021, Daunhawer et al., 2023, Rusak et al., 2024]. We corroborate our theoretical claims on synthetic data adhering to our assumptions—we even show that good performance is possible when the assumptions are violated. Notably, we observe that cluster centroids recovery from DIET's classification head is more robust than ground-truth representation prediction from the learned representation.

## 2 Identifiability guarantees for DIET

This section presents our main theoretical contribution. After summarizing DIET, we introduce a mildly constrained theoretical setup, in which DIET provably recovers the correct latents. The setup is followed by the main result and a discussion on the intuition for our theoretical model.

**DIET [Ibrahim et al., 2024].** DIET solves an instance classification problem, where each sample $\boldsymbol{x}$ in the training dataset has a unique instance label $i$. Augmentations do not affect this label. We have a composite model $\boldsymbol{W} \circ \boldsymbol{f}$, where the backbone $\boldsymbol{f}$ produces $d$-dimensional representations, and a linear, bias-free classification head $\boldsymbol{W}$ that maps these representations to a logit vector equal in size to the cardinality of the training dataset. If the parameter vector corresponding to logit $i$ is denoted as $\boldsymbol{w}_i$, then $\boldsymbol{W}$ effectively computes similarity scores (scalar products) between the $\boldsymbol{w}_i$'s and embeddings $\boldsymbol{f}(\boldsymbol{x})$. DIET trains this architecture to predict the correct instance label using multinomial regression (with $\boldsymbol{f}, \boldsymbol{W}$ and temperature $\beta$ as variables):

$$\mathcal{L}(\boldsymbol{f}, \boldsymbol{W}, \beta) = \mathbb{E}_{(\boldsymbol{x},i)} \left[ -\ln \frac{e^{\beta \langle \boldsymbol{w}_i, \boldsymbol{f}(\boldsymbol{x}) \rangle}}{\sum_j e^{\beta \langle \boldsymbol{w}_j, \boldsymbol{f}(\boldsymbol{x}) \rangle}} \right]. \tag{1}$$

**Setup.** For our theory, we need to formally define a latent variable model (LVM) for the data generating process (DGP) to assess the identifiability of latent factors. For this, we take a cluster-centric approach, representing semantic classes by cluster vectors, similar to proxy-based metric learning [Kirchhof et al., 2022]. Then, we model the samples of a class with a von Mises-Fisher (vMF) distribution, centered around the class's cluster vector. This conditional distribution jointly models intra-class sample selection and *augmentations* of samples, together called *intra-class variances*. Our conditional does not mean that each sample pair transforms into each other via augmentations *with high probability*. It does mean that—since we assume an LVM on the hypersphere; i.e., all

semantic concepts (color, position, etc.) correspond to a continuous latent factor—the latent manifold is connected, or equivalently, that the augmentation graph is connected, which is an assumption used in [Wang et al., 2022, Balestriero and LeCun, 2022, HaoChen et al., 2022]. We provide an overview of our assumptions, and defer additional details to Assums. 1C in Appx. A:

**Assumptions 1** (DGP with vMF samples around cluster vectors. *Details omitted.*)**.**

  (i) *There is a finite set of semantic classes $\mathscr{C}$, represented by a set of unit-norm $d$-dimensional cluster-vectors $\{\boldsymbol{v}_c | c \in \mathscr{C}\} \subseteq \mathbb{S}^{d-1}$. The system $\{\boldsymbol{v}_c\}$ is sufficiently large and spread out.*

  (ii) *Any sample $i$ belongs to exactly one class $c = \mathcal{C}(i)$.*

  (iii) *The latent $\boldsymbol{z} \in \mathbb{S}^{d-1}$ of our data sample with instance label $i$ is drawn from a vMF distribution around the cluster vector $\boldsymbol{v}_c$ of class $c = \mathcal{C}(i)$:*

$$\boldsymbol{z} \sim p(\boldsymbol{z}|c) \propto e^{\alpha \langle \boldsymbol{v}_c, \boldsymbol{z} \rangle}. \tag{2}$$

  (iv) *Sample $\boldsymbol{x}$ is generated by passing latent $\boldsymbol{z}$ through an injective generator function: $\boldsymbol{x} = \boldsymbol{g}(\boldsymbol{z})$.*

**Main result.** Under Assums. 1, we prove the identifiability of both the latent representations and the cluster vectors, $\boldsymbol{v}_c$, in all four combinations of unit-normalized (i.e., when the latent space is the hypersphere, commonly used, e.g., in InfoNCE [Chen et al., 2020]); and non-normalized (as in the original DIET paper [Ibrahim et al., 2024]) latents, $\boldsymbol{z}$, and weight vectors, $\boldsymbol{w}_i$ . We state a concise version of our result and defer the full treatment and the proof to Thm. 1C in Appx. A:

**Theorem 1** (Identifiability of latents drawn from vMF around cluster vectors. *Details omitted.*)**.** *Let $(\boldsymbol{f}, \boldsymbol{W}, \beta)$ globally minimize the DIET objective (1) under the following additional constraints:*

C3. *the embeddings $\boldsymbol{f}(\boldsymbol{x})$ are unnormalized, while the $\boldsymbol{w}_i$'s are unit-normalized. Then $\boldsymbol{w}_i$ identifies the cluster vector $\boldsymbol{v}_{\mathcal{C}(i)}$ up to an orthogonal linear transformation $\mathcal{O}$: $\boldsymbol{w}_i = \mathcal{O}\boldsymbol{v}_{\mathcal{C}(i)}$, for any $i$. Furthermore, the inferred latents $\tilde{z} = \boldsymbol{f}(\boldsymbol{x})$ identify the ground-truth latents $\boldsymbol{z}$ up to the same orthogonal transformation, but scaled.*

C4. *neither the embeddings $\boldsymbol{f}(\boldsymbol{x})$ nor the $\boldsymbol{w}_i$'s are unit-normalized. Then the cluster vectors $\boldsymbol{v}_c$ and the latent $\boldsymbol{z}$ are identified up to an affine linear and linear transformation, respectively.*

*In all cases, the weight vectors belonging to samples of the same class are equal, i.e., for any $i, j$, $\mathcal{C}(i) = \mathcal{C}(j)$ implies $\boldsymbol{w}_i = \boldsymbol{w}_j$.*

**Intuition.** DIET assigns a different (instance) label and a unique weight vector $\boldsymbol{w}_i$ to each training sample. The cross-entropy objective is optimized if the trained neural network can distinguish between the samples. Thus, the learned representation $\tilde{z} = \boldsymbol{f}(\boldsymbol{x})$ should capture enough information to distinguish different samples, even from the same class.

However, the weight vectors $\boldsymbol{w}_i$'s cannot be sensitive to the intra-class sample variance or the sample's instance label $i$ (because multiple instances will usually belong to the same class). This leads to the weight vectors taking the values of the cluster vectors. As cluster vectors only capture some statistics of the conditional, feature recovery is more fine-grained than cluster identifiability. The interaction between the two is dictated by the cross-entropy loss, which is minimized if the representation $\tilde{z}$ is most similar to its own assigned weight vector $\boldsymbol{w}_i$. Fig. 1 provides a visualization conveying the intuition behind Thm. 1.

## 3  Experiments

In the following section, we empirically verify the claims made in Thm. 1 in the synthetic setting. We generate data samples according to Assums. 1: ground-truth latents are sampled around cluster centroids $\boldsymbol{v}_c$ following a vMF distribution. Data augmentations, which share the same instance label $i$, are sampled from the same vMF distribution around $\boldsymbol{v}_c$.

**Synthetic Setup.** We consider $N$ data samples of dimensionality $d$ generated from $\boldsymbol{z} \sim p(\boldsymbol{z}|\mathbf{v}_c)$, sampled around a set of $|\mathscr{C}|$ class vectors, $\boldsymbol{v}_c$ uniformly distributed across the unit hyper-sphere. We use an invertible multi-layer perceptron (MLP) to map ground truth latents to data samples. We train a classification head $\boldsymbol{W} = [\boldsymbol{w}_i^\top |_{i=1}^N]$ and an MLP encoder that maps samples to representations $\tilde{z} \in \mathbb{R}^d$ using the DIET objective (1). While to verify Thm. 1 case C4., we do not normalize $\boldsymbol{W}$, we do unit-normalize the weight vectors to validate Thm. 1 case C3. We verify our theoretical claims by measuring the predictability of the ground-truth $\boldsymbol{z}$ from $\tilde{z}$ and $\boldsymbol{v}_c$ from $\boldsymbol{w}_i$ using the $R^2$ score on a held-out dataset. For identifiability up to orthogonal linear transformations, we train linear mappings

with no intercept, assess the $R^2$ score and verify that the singular values of this transformation converge to one, while for identifiability up to affine linear transformations, we simply assess the predictive accuracy of a linear predictor with intercept.

Table 1: Identifiability in the synthetic setup. Mean $\pm$ standard deviation across 5 random seeds. Settings that match and violate our theoretical assumptions are ✓ and ✗ respectively. We report the $R^2$ score for linear mappings, $\tilde{z} \to z$ and $w_i \to v_c$ for cases with normalized (o) and not normalized (a) $w_i$. For normalized $w_i$, we verify that mappings $\tilde{z} \to z$ are orthogonal by reporting the mean absolute error between their singular values and those of an orthogonal transformation.

| | | | | | normalized $w_i$ cases | | | | unnormalized $w_i$ | |
| | | | | | $R_o^2(\uparrow)$ | | $\mathrm{MAE_o}(\downarrow)$ | | $R_a^2(\uparrow)$ | |
| $N$ | $d$ | $\lvert\mathscr{C}\rvert$ | $p(z\lvert v_c)$ | M. | $\tilde{z} \to z$ | $w_i \to v_c$ | $\tilde{z} \to z$ | $w_i \to v_c$ | $\tilde{z} \to z$ | $w_i \to v_c$ |
|---|---|---|---|---|---|---|---|---|---|---|
| $10^3$ | 5 | 100 | vMF($\kappa=10$) | ✓ | $98.6_{\pm0.01}$ | $99.9_{\pm0.00}$ | $0.01_{\pm0.00}$ | $0.00_{\pm0.00}$ | $99.0_{\pm0.00}$ | $99.9_{\pm0.00}$ |
| $10^5$ | 5 | 100 | vMF($\kappa=10$) | ✓ | $98.2_{\pm0.01}$ | $99.5_{\pm0.00}$ | $0.00_{\pm0.00}$ | $0.00_{\pm0.00}$ | $99.7_{\pm0.00}$ | $99.8_{\pm0.00}$ |
| $10^3$ | 5 | 100 | vMF($\kappa=10$) | ✓ | $98.6_{\pm0.01}$ | $99.9_{\pm0.00}$ | $0.01_{\pm0.00}$ | $0.00_{\pm0.00}$ | $99.0_{\pm0.00}$ | $99.9_{\pm0.00}$ |
| $10^3$ | 10 | 100 | vMF($\kappa=10$) | ✓ | $92.5_{\pm0.01}$ | $99.6_{\pm0.00}$ | $0.01_{\pm0.00}$ | $0.00_{\pm0.00}$ | $93.0_{\pm0.03}$ | $99.6_{\pm0.00}$ |
| $10^3$ | 20 | 100 | vMF($\kappa=10$) | ✓ | $70.8_{\pm0.02}$ | $97.1_{\pm0.01}$ | $0.03_{\pm0.00}$ | $0.00_{\pm0.00}$ | $81.9_{\pm0.01}$ | $99.7_{\pm0.00}$ |
| $10^3$ | 5 | 10 | vMF($\kappa=10$) | ✓ | $88.6_{\pm0.05}$ | $85.7_{\pm0.15}$ | $0.02_{\pm0.00}$ | $0.00_{\pm0.00}$ | $90.0_{\pm0.05}$ | $99.0_{\pm0.03}$ |
| $10^3$ | 5 | 100 | vMF($\kappa=10$) | ✓ | $98.6_{\pm0.01}$ | $99.9_{\pm0.01}$ | $0.01_{\pm0.00}$ | $0.00_{\pm0.00}$ | $99.0_{\pm0.00}$ | $99.9_{\pm0.00}$ |
| $10^3$ | 5 | 1000 | vMF($\kappa=10$) | ✓ | $99.3_{\pm0.00}$ | $99.9_{\pm0.00}$ | $0.00_{\pm0.00}$ | $0.00_{\pm0.00}$ | $99.2_{\pm0.00}$ | $99.9_{\pm0.00}$ |
| $10^3$ | 5 | 100 | vMF($\kappa=5$) | ✓ | $98.6_{\pm0.01}$ | $99.9_{\pm0.01}$ | $0.01_{\pm0.00}$ | $0.00_{\pm0.00}$ | $99.0_{\pm0.00}$ | $99.8_{\pm0.00}$ |
| $10^3$ | 5 | 100 | vMF($\kappa=10$) | ✓ | $99.0_{\pm0.00}$ | $99.9_{\pm0.00}$ | $0.00_{\pm0.00}$ | $0.00_{\pm0.00}$ | $99.1_{\pm0.00}$ | $99.9_{\pm0.00}$ |
| $10^3$ | 5 | 100 | vMF($\kappa=50$) | ✓ | $45.0_{\pm0.06}$ | $49.7_{\pm0.06}$ | $0.30_{\pm0.00}$ | $0.00_{\pm0.00}$ | $72.5_{\pm0.03}$ | $75.5_{\pm0.00}$ |
| $10^3$ | 5 | 100 | vMF($\kappa=10$) | ✓ | $98.6_{\pm0.01}$ | $99.9_{\pm0.01}$ | $0.01_{\pm0.00}$ | $0.00_{\pm0.00}$ | $99.0_{\pm0.00}$ | $99.9_{\pm0.00}$ |
| $10^3$ | 5 | 100 | Laplace ($b=1.0$) | ✗ | $85.2_{\pm0.01}$ | $99.7_{\pm0.01}$ | $0.01_{\pm0.00}$ | $0.00_{\pm0.00}$ | $85.4_{\pm0.00}$ | $99.5_{\pm0.00}$ |
| $10^3$ | 5 | 100 | Normal ($\sigma^2=1.0$) | ✗ | $98.7_{\pm0.00}$ | $99.8_{\pm0.00}$ | $0.01_{\pm0.00}$ | $0.00_{\pm0.00}$ | $98.6_{\pm0.00}$ | $99.6_{\pm0.00}$ |

**Results.** Tab. 1 depicts our results for synthetic experiments. For both cases, when $W$ is and is not unit-normalized, the $R^2$ score for both the latents and the cluster vectors is close to $100\%$, except when the latent dimensionality is 20—such scalability problems are a common artifact in SSL [Zimmermann et al., 2021, Rusak et al., 2024]. For unit-normalized $W$, the MAE is close to zero even in such cases. For a higher concentration of samples around $v_c$ (i.e., $\kappa=50$) as well as a lower number of clusters (i.e., $\lvert\mathscr{C}\rvert=10$), the $R^2$ score decreases, which is also a common phenomenon, and is possibly explained by too strong augmentation overlap [Wang et al., 2022, Rusak et al., 2024]. For a low number of clusters, high $\kappa$ and a fixed number of training samples, the concentration of samples in regions surrounding centroids, $v_c$, increases, a setting, refered to as "overly overlapping augmentations", known to be suboptimal and leading to a drop in downstream performance [Wang et al., 2022]. Our results also suggest that even under model misspecification (last two rows with non-vMF latent distributions), identifiability still holds. We provide an additional ablation study for the concentration of $v_c$ across the unit hyper-sphere in Appx. B.

## 4 Discussion

**Limitations.** Our analysis proves the identifiability of DIET [Ibrahim et al., 2024] with a cluster-based DGP, thus providing the first such result for self-supervised parametric instance classification methods. However, our theory cannot yet explain the importance of label smoothing in DIET, noted by Ibrahim et al. [2024], and it also remains to be seen whether such identifiability results scale for larger datasets, for which the large-dimensional classifier head in DIET in the original form is prohibitive. It also remains an issue that the vMF conditional distribution around cluster centroids jointly models intra-class sample selection and augmentations of samples, as we suspect that the supports of augmentation spaces of different samples do not overlap as much as it would be suggested by the choice of conditional. Also, we leave it for future work to investigate a formal connection to nonlinear ICA methods such as InfoNCE [Zimmermann et al., 2021] or the Generalized Contrastive Learning framework [Hyvarinen et al., 2019].

**Conclusion.** By modeling the DGP in DIET [Ibrahim et al., 2024] with a cluster-based latent variable model, we provide identifiability results for both the latent representation and the cluster vectors, which is the first of its kind for self-supervised instance discrimination methods. We also showcase this in synthetic settings, where we recover both the latents and cluster vectors even under model misspecification. We hope that our work inspires further research into investigating the theoretical guarantees of simplified but effective SSL methods like DIET.

## Acknowledgments

The authors thank the International Max Planck Research School for Intelligent Systems (IMPRS-IS) for supporting Patrik Reizinger and Attila Juhos. Patrik Reizinger acknowledges his membership in the European Laboratory for Learning and Intelligent Systems (ELLIS) PhD program. This work was supported by the German Federal Ministry of Education and Research (BMBF): Tübingen AI Center, FKZ: 01IS18039A. Wieland Brendel acknowledges financial support via an Emmy Noether Grant funded by the German Research Foundation (DFG) under grant no. BR 6382/1-1 and via the Open Philantropy Foundation funded by the Good Ventures Foundation. Wieland Brendel is a member of the Machine Learning Cluster of Excellence, EXC number 2064/1 – Project number 390727645. This research utilized compute resources at the Tübingen Machine Learning Cloud, DFG FKZ INST 37/1057-1 FUGG. Alice Bizeul's work is supported by an ETH AI Center Doctoral fellowship.

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

# A Identifiability of latents drawn from a vMF around cluster vectors

In this section, we formally state and prove our core theoretical result. We start off by defining and discussing a useful notion, then introduce our assumptions on the data generating process. We proceed with the main statement and finish with the proof.

## A.1 Affine Generator Systems

**Definition 1** (Affine Generator System). *A system of vectors $\{\boldsymbol{v}_c \in \mathbb{R}^d | c \in \mathscr{C}\}$ is called an affine generator system if the affine hull defined by them is $\mathbb{R}^d$. More precisely, any vector in $\mathbb{R}^d$ is an affine linear combination of the vectors in the system. Put into symbols: for any $\boldsymbol{v} \in \mathbb{R}^d$ there exist coefficients $\alpha_c \in \mathbb{R}$, such that*

$$\boldsymbol{v} = \sum_{c \in \mathscr{C}} \alpha_c \boldsymbol{v}_c \quad \text{and} \quad \sum_{c \in \mathscr{C}} \alpha_c = 1. \tag{3}$$

**Lemma 1** (Properties of affine generator systems). *The following hold for any affine generator system $\{\boldsymbol{v}_c \in \mathbb{R}^d | c \in \mathscr{C}\}$:*

*1. for any $a \in \mathscr{C}$ the system $\{\boldsymbol{v}_c - \boldsymbol{v}_a | c \in \mathscr{C}\}$ is now a generator system of $\mathbb{R}^d$;*
*2. the invertible linear image of an affine generator system is also an affine generator system.*

## A.2 Assumptions and main result

**Assumptions 1C** (DGP with vMF samples around cluster vectors). *Assume the following DGP:*

- *(i) There exists a finite set of classes $\mathscr{C}$, represented by a set of unit-norm $d$-dimensional cluster-vectors $\{\boldsymbol{v}_c | c \in \mathscr{C}\} \subseteq \mathbb{S}^{d-1}$ such that they form an affine generator system of $\mathbb{R}^d$.*
- *(ii) There is a finite set of instace labels $\mathscr{I}$ and a well-defined, surjective class function $\mathcal{C} : \mathscr{I} \to \mathscr{C}$ (every label belongs to exactly one class and every class is in use).*
- *(iii) Our data sample is labelled with an instance label chosen uniformly, i.e., $I \in Uni(\mathscr{I})$ and, hence, belongs to class $C = \mathcal{C}(I)$.*
- *(iv) The latent $\boldsymbol{z} \in \mathbb{S}^{d-1}$ of our data sample with label $I$ is drawn from a vMF distribution around the cluster vector $\boldsymbol{v}_C$, where $C = \mathcal{C}(I)$:*

$$\boldsymbol{z} \sim p(\boldsymbol{z}|C) \propto e^{\alpha \langle \boldsymbol{v}_C, \boldsymbol{z} \rangle}. \tag{4}$$

- *(v) The data sample $\boldsymbol{x}$ is generated by passing the latent $\boldsymbol{z}$ through a continuous and injective generator function $\boldsymbol{g} : \mathbb{S}^{d-1} \to \mathbb{R}^D$, i.e., $\boldsymbol{x} = \boldsymbol{g}(\boldsymbol{z})$.*

Assume that, using the DIET objective (6), we train a continuous encoder $\boldsymbol{f} : \mathbb{R}^D \to \mathbb{R}^d$ on $\boldsymbol{x}$ and a linear classification head $\boldsymbol{W}$ on top of $\boldsymbol{f}$. The rows of $\boldsymbol{W}$ are $\{\boldsymbol{w}_i^\top | i \in \mathscr{I}\}$. In other words, $\boldsymbol{W}$ computes similarities (scalar products) between its rows and the embeddings:

$$\boldsymbol{W} : \boldsymbol{f}(\boldsymbol{x}) \mapsto \left[ \langle \boldsymbol{w}_i, \boldsymbol{f}(\boldsymbol{x}) \rangle |_{i \in \mathscr{I}} \right]. \tag{5}$$

In DIET, we optimize the following objective amongst all possible continuous encoders $\boldsymbol{f}$, linear classifiers $\boldsymbol{W}$, and $\beta > 0$:

$$\mathcal{L}(\boldsymbol{f}, \boldsymbol{W}, \beta) = \mathbb{E}_{(\boldsymbol{x}, I)} \left[ -\ln \frac{e^{\beta \langle \boldsymbol{w}_I, \boldsymbol{f}(\boldsymbol{x}) \rangle}}{\sum_{j \in \mathscr{I}} e^{\beta \langle \boldsymbol{w}_j, \boldsymbol{f}(\boldsymbol{x}) \rangle}} \right] \tag{6}$$

**Theorem 1C** (Identifiability of latents drawn from a vMF around cluster vectors). *Let $(\boldsymbol{f}, \boldsymbol{W}, \beta)$ globally minimize the DIET objective (6) under the following additional constraints:*

*C1. both the embeddings $\boldsymbol{f}(\boldsymbol{x})$ and $\boldsymbol{w}_i$'s are unit-normalized. Then:*
- *(a) $\boldsymbol{h} = \boldsymbol{f} \circ \boldsymbol{g}$ is orthogonal linear, i.e., the latents are identified up to an orthogonal linear transformation;*
- *(b) $\boldsymbol{w}_i = \boldsymbol{h}(\boldsymbol{v}_{\mathcal{C}(i)})$ for any $i \in \mathscr{I}$, i.e., $\boldsymbol{w}_i$'s identify the cluster-vectors $\boldsymbol{v}_c$ up to the same orthogonal linear transformation;*
- *(c) $\beta = \alpha$, the temperature of the vMF distribution is also identified.*

*C2. the embeddings $\boldsymbol{f}(\boldsymbol{x})$ are unit-normalized, the $\boldsymbol{w}_i$'s are unnormalized. Then:*
- *(a) $\boldsymbol{h} = \boldsymbol{f} \circ \boldsymbol{g}$ is orthogonal linear;*
- *(b) $\boldsymbol{w}_i = \frac{\alpha}{\beta} \boldsymbol{h}(\boldsymbol{v}_{\mathcal{C}(i)}) + \boldsymbol{\psi}$ for any $i \in \mathscr{I}$, where $\boldsymbol{\psi}$ is a constant vector independent of $i$.*

*C3. the embeddings $\boldsymbol{f}(\boldsymbol{x})$ are unnormalized, while the $\boldsymbol{w}_i$'s are unit-normalized. If the system $\{\boldsymbol{v}_c|c\}$ **is diverse enough in the sense of** Assum. 2, then:*

    *(a) $\boldsymbol{w}_i = \mathcal{O}\boldsymbol{v}_{\mathcal{C}(i)}$, for any $i \in \mathscr{I}$, where $\mathcal{O}$ is orthogonal linear;*

    *(b) $\boldsymbol{h} = \boldsymbol{f} \circ \boldsymbol{g} = \frac{\alpha}{\beta}\mathcal{O}$ with the same orthogonal linear transformation, but scaled with $\frac{\alpha}{\beta}$.*

*C4. neither the embeddings $\boldsymbol{f}(\boldsymbol{x})$ nor the rows of $\boldsymbol{W}$ are unit-normalized. Then:*

    *(a) $\boldsymbol{h} = \boldsymbol{f} \circ \boldsymbol{g}$ is linear;*

    *(b) $\boldsymbol{w}_i$ identifies $\boldsymbol{v}_{\mathcal{C}(i)}$ up to an affine linear transformation.*

*Furthermore, in all cases, the row vectors that belong to samples of the same class are equal, i.e., for any $i, j \in \mathscr{I}$, $\mathcal{C}(i) = \mathcal{C}(j)$ implies $\boldsymbol{w}_i = \boldsymbol{w}_j$.*

**Remark.** *In cases C2 and C4, the cluster vectors are unnormalized and, therefore, can absorb the temperature parameter $\beta$. Thus $\beta$ can be set to $1$ without loss of generality. In case C3, it is $\boldsymbol{f}$ that can absorb $\beta$.*

**Assumption 2** (Diverse data). *The system $\{\boldsymbol{v}_c|c \in \mathscr{C}\}$ is said to be diverse enough, if the following $|\mathscr{C}| \times 2d$ matrix has full column rank of $2d$:*

$$\begin{pmatrix} \cdots\cdots\cdots & \cdots\cdots\cdots \\ (\boldsymbol{v}_c \odot \boldsymbol{v}_c)^\top & \boldsymbol{v}_c^\top \\ \cdots\cdots\cdots & \cdots\cdots\cdots \end{pmatrix}, \tag{7}$$

*where $[\boldsymbol{x} \odot \boldsymbol{y}]_i = x_i y_i$ is the elementwise- or Hadamard product.*

*As long as $|\mathscr{C}| \geq 2d$, this property holds almost surely w.r.t. the Lebesgue-measure of $\mathbb{S}^{d-1}$ or any continuous probability distribution of $\boldsymbol{v}_c \in \mathbb{S}^{d-1}$.*

*Proof.* **Step 1: Deriving an equation characterizing the global optimizers of the objective.**

**Rewriting the objective in terms of latents:**    we plug the expression $\boldsymbol{x} = \boldsymbol{g}(\boldsymbol{z})$ into the optimization objective (6) to express the dependence in terms of the latents $\boldsymbol{z}$:

$$\mathcal{L}(\boldsymbol{f}, \boldsymbol{W}, \beta) = \mathbb{E}_{(\boldsymbol{z}, I)}\left[-\ln\frac{e^{\beta\langle\boldsymbol{w}_I, \boldsymbol{f}\circ\boldsymbol{g}(\boldsymbol{z})\rangle}}{\sum_{j\in\mathscr{I}} e^{\beta\langle\boldsymbol{w}_j, \boldsymbol{f}\circ\boldsymbol{g}(\boldsymbol{z})\rangle}}\right] = \mathcal{L}_{\boldsymbol{z}}(\boldsymbol{f}\circ\boldsymbol{g}, \boldsymbol{W}, \beta), \tag{8}$$

where the optimization is still over $\boldsymbol{f}$ (and not $\boldsymbol{h} = \boldsymbol{f} \circ \boldsymbol{g}$).

We note that the generator $\boldsymbol{g}$ is, by assumption, continuously invertible on the *compact* set $\mathbb{S}^{d-1}$. Therefore, its image $\boldsymbol{g}(\mathbb{S}^{d-1})$ is compact, too, and its inverse $\boldsymbol{g}^{-1}$ is also continuous. By Tietze's extension theorem [Wikipedia, 2024b], $\boldsymbol{g}^{-1}$ can be continuously extended to a function $\boldsymbol{F} : \mathbb{R}^D \to \mathbb{S}^{d-1}$. Therefore, any continuous function $\boldsymbol{h} : \mathbb{S}^{d-1} \to \mathbb{R}^d$ can take the role of $\boldsymbol{f} \circ \boldsymbol{g}$ by substituting $\boldsymbol{f} = \boldsymbol{h} \circ \boldsymbol{F}$ continuous, since now $\boldsymbol{f} \circ \boldsymbol{g} = \boldsymbol{h} \circ (\boldsymbol{F} \circ \boldsymbol{g}) = \boldsymbol{h} \circ id_{\mathbb{S}^{d-1}} = \boldsymbol{h}$.

Hence, minimizing $\mathcal{L}_{\boldsymbol{z}}(\boldsymbol{f} \circ \boldsymbol{g}, \boldsymbol{W}, \beta)$ (and by extension $\mathcal{L}(\boldsymbol{f}, \boldsymbol{W}, \beta)$) for continuous $\boldsymbol{f}$ equates to minimizing $\mathcal{L}_{\boldsymbol{z}}(\boldsymbol{h}, \boldsymbol{W}, \beta)$ for continuous $\boldsymbol{h}$:

$$\mathcal{L}_{\boldsymbol{z}}(\boldsymbol{h}, \boldsymbol{W}, \beta) = \mathbb{E}_{(\boldsymbol{z}, I)}\left[-\ln\frac{e^{\beta\langle\boldsymbol{w}_I, \boldsymbol{h}(\boldsymbol{z})\rangle}}{\sum_{j\in\mathscr{I}} e^{\beta\langle\boldsymbol{w}_j, \boldsymbol{h}(\boldsymbol{z})\rangle}}\right]. \tag{9}$$

**Expressing the condition for global optimality of the objective:**    We rewrite the objective (9) by 1) using the indicator variable $\delta_{I=i}$ of the event $\{I = i\}$ and 2) applying the law of total expectation:

$$\mathcal{L}_{\boldsymbol{z}}(\boldsymbol{h}, \boldsymbol{W}, \beta) = \mathbb{E}_{(\boldsymbol{z}, I)}\left[-\sum_{i\in\mathscr{I}}\delta_{I=i}\ln\frac{e^{\beta\langle\boldsymbol{w}_i, \boldsymbol{h}(\boldsymbol{z})\rangle}}{\sum_{j\in\mathscr{I}} e^{\beta\langle\boldsymbol{w}_j, \boldsymbol{h}(\boldsymbol{z})\rangle}}\right] \tag{10}$$

$$= \mathbb{E}_{\boldsymbol{z}}\left[\mathbb{E}_I\left[-\sum_{i\in\mathscr{I}}\delta_{I=i}\ln\frac{e^{\beta\langle\boldsymbol{w}_i, \boldsymbol{h}(\boldsymbol{z})\rangle}}{\sum_{j\in\mathscr{I}} e^{\beta\langle\boldsymbol{w}_j, \boldsymbol{h}(\boldsymbol{z})\rangle}}\,\Big|\,\boldsymbol{z}\right]\right]. \tag{11}$$

Using the properties that $\mathbb{E}\big[A\,f(B)\big|B\big] = \mathbb{E}\big[A\big|B\big]f(B)$ and that $\mathbb{E}[\delta_{I=i}] = \mathbb{P}(I = i)$, we conclude that:

$$\mathcal{L}_{\boldsymbol{z}}(\boldsymbol{h}, \boldsymbol{W}, \beta) = \mathbb{E}_{\boldsymbol{z}}\left[-\sum_{i\in\mathscr{I}}\mathbb{E}_I\left[\delta_{I=i}\ln\frac{e^{\beta\langle\boldsymbol{w}_i,\boldsymbol{h}(\boldsymbol{z})\rangle}}{\sum_{j\in\mathscr{I}}e^{\beta\langle\boldsymbol{w}_j,\boldsymbol{h}(\boldsymbol{z})\rangle}}\,\Big|\,\boldsymbol{z}\right]\right] \tag{12}$$

$$= \mathbb{E}_{\boldsymbol{z}}\left[-\sum_{i\in\mathscr{I}}\mathbb{E}_I\left[\delta_{I=i}\big|\boldsymbol{z}\right]\ln\frac{e^{\beta\langle\boldsymbol{w}_i,\boldsymbol{h}(\boldsymbol{z})\rangle}}{\sum_{j\in\mathscr{I}}e^{\beta\langle\boldsymbol{w}_j,\boldsymbol{h}(\boldsymbol{z})\rangle}}\right] \tag{13}$$

$$= \mathbb{E}_{\boldsymbol{z}}\left[-\sum_{i\in\mathscr{I}}\mathbb{P}(I = i|\boldsymbol{z})\ln\frac{e^{\beta\langle\boldsymbol{w}_i,\boldsymbol{h}(\boldsymbol{z})\rangle}}{\sum_{j\in\mathscr{I}}e^{\beta\langle\boldsymbol{w}_j,\boldsymbol{h}(\boldsymbol{z})\rangle}}\right]. \tag{14}$$

By Gibbs' inequality [Wikipedia, 2024a], the cross-entropy inside the expectation is globally minimized if and only if

$$\frac{e^{\beta\langle\boldsymbol{w}_i,\boldsymbol{h}(\boldsymbol{z})\rangle}}{\sum_{j\in\mathscr{I}}e^{\beta\langle\boldsymbol{w}_j,\boldsymbol{h}(\boldsymbol{z})\rangle}} = \mathbb{P}(I = i|\boldsymbol{z}), \quad \text{for any } i\in\mathscr{I}. \tag{15}$$

Moreover, the entire expectation is globally minimized if and only if the above equality (15) holds almost everywhere for $\boldsymbol{z}\in\mathbb{S}^{d-1}$.

Using that instance label $I$ is uniformly distributed, or $\mathbb{P}(I = j) = \mathbb{P}(I = i)$, the likelihood of the sample being in class $i$ can be expressed via Bayes' theorem as:

$$\mathbb{P}(I = i|\boldsymbol{z}) = \frac{p(\boldsymbol{z}|I = i)\mathbb{P}(I = i)}{\sum_{j\in\mathscr{I}}p(\boldsymbol{z}|I = j)\mathbb{P}(I = j)} = \frac{p(\boldsymbol{z}|I = i)}{\sum_{j\in\mathscr{I}}p(\boldsymbol{z}|I = j)}. \tag{16}$$

Substituting (16) into (15) yields that for any $i\in\mathscr{I}$ and almost everywhere w.r.t. $\boldsymbol{z}\in\mathbb{S}^{d-1}$:

$$\frac{e^{\beta\langle\boldsymbol{w}_i,\boldsymbol{h}(\boldsymbol{z})\rangle}}{\sum_{j\in\mathscr{I}}e^{\beta\langle\boldsymbol{w}_j,\boldsymbol{h}(\boldsymbol{z})\rangle}} = \frac{p(\boldsymbol{z}|I = i)}{\sum_{j\in\mathscr{I}}p(\boldsymbol{z}|I = j)}. \tag{17}$$

We now divide the equation (17) for the probability of a sample having label $i$ with that of having label $k$ and take the logarithm. This yields that $\mathcal{L}_{\boldsymbol{z}}(\boldsymbol{h}, \boldsymbol{W}, \beta)$ is globally minimized if and only if

$$\beta\langle\boldsymbol{w}_i - \boldsymbol{w}_k, \boldsymbol{h}(\boldsymbol{z})\rangle = \ln\frac{p(\boldsymbol{z}|I = i)}{p(\boldsymbol{z}|I = k)} \tag{18}$$

holds for any $i, k\in\mathscr{I}$ and almost everywhere w.r.t. $\boldsymbol{z}\in\mathbb{S}^{d-1}$.

**Plugging in the vMF distribution:** Plugging the assumed conditional distribution from (4) into (18) yields the equivalent expression:

$$\beta\langle\boldsymbol{w}_i - \boldsymbol{w}_k, \boldsymbol{h}(\boldsymbol{z})\rangle = \alpha\langle\boldsymbol{v}_{\mathcal{C}(i)} - \boldsymbol{v}_{\mathcal{C}(k)}, \boldsymbol{z}\rangle \tag{19}$$

holds for any $i, k\in\mathscr{I}$ and almost everywhere w.r.t. $\boldsymbol{z}\in\mathbb{S}^{d-1}$. Since $\boldsymbol{h}$ is continuous, the equation holds almost everywhere w.r.t. $\boldsymbol{z}$ if and only if it holds for all $\boldsymbol{z}\in\mathbb{S}^{d-1}$.

Observe that if $\boldsymbol{h} = id|_{\mathbb{S}^{d-1}}$, $\boldsymbol{w}_i = \boldsymbol{v}_{\mathcal{C}(i)}$ for any $i\in\mathscr{I}$, and $\beta = \alpha$, then the equation is satisfied. Thus, we can conclude that the global minimum of the cross-entropy loss is achieved.

**Step 2: Solving the equation for $h, W$ and proving identifiability.**

We now find all solutions to prove the identifiability of the latent variables and that of the cluster vectors. Denote $\tilde{\boldsymbol{w}}_i = \frac{\beta}{\alpha}\boldsymbol{w}_i$ to simplify the above equation to:

$$\langle\tilde{\boldsymbol{w}}_i - \tilde{\boldsymbol{w}}_k, \boldsymbol{h}(\boldsymbol{z})\rangle = \langle\boldsymbol{v}_{\mathcal{C}(i)} - \boldsymbol{v}_{\mathcal{C}(k)}, \boldsymbol{z}\rangle. \tag{20}$$

**$h$ is injective and has full-dimensional image:** We prove that $\boldsymbol{h}$ is injective. Assume that $\boldsymbol{h}(\boldsymbol{z}_1) = \boldsymbol{h}(\boldsymbol{z}_2)$ for some $\boldsymbol{z}_1, \boldsymbol{z}_2\in\mathbb{S}^{d-1}$. Plugging $\boldsymbol{z}_1$ and $\boldsymbol{z}_2$ into (20) and subtracting the two equations yields:

$$0 = \langle\tilde{\boldsymbol{w}}_i - \tilde{\boldsymbol{w}}_k, \boldsymbol{h}(\boldsymbol{z}_1) - \boldsymbol{h}(\boldsymbol{z}_2)\rangle = \langle\boldsymbol{v}_{\mathcal{C}(i)} - \boldsymbol{v}_{\mathcal{C}(k)}, \boldsymbol{z}_1 - \boldsymbol{z}_2\rangle, \tag{21}$$

for any $i, k$. However, as the cluster vectors $\{\boldsymbol{v}_c|c\}$ form an affine generator system, the vectors $\{\boldsymbol{v}_{\mathcal{C}(i)} - \boldsymbol{v}_{\mathcal{C}(k)}|i, k\}$ form a generator system of $\mathbb{R}^d$ (see Lem. 1). Therefore, $\langle\boldsymbol{y}, \boldsymbol{z}_1 - \boldsymbol{z}_2\rangle = 0$, for any $\boldsymbol{y}\in\mathbb{R}^d$, which holds if and only if $\boldsymbol{z}_1 = \boldsymbol{z}_2$. Hence, $\boldsymbol{h}$ is injective.

By the Borsuk-Ulam theorem, for any continuous map from $\mathbb{S}^{d-1}$ to a space of dimensionality at most $d-1$ there exists some pair of antipodal points that are mapped to the same point. Consequently, no such function can be injective at the same time. Since $h : \mathbb{S}^{d-1}\to\mathbb{R}^d$ is injective, the linear span of its image must be $\mathbb{R}^d$.

**Collapse of $w_i$'s:**  We prove that $\tilde{w}_i = \tilde{w}_k$ if $\mathcal{C}(i) = \mathcal{C}(k)$, i.e., samples from the same cluster will have equal rows of $W$ associated with them.

Assume that $\mathcal{C}(i) = \mathcal{C}(k)$ and substitute them into (20):

$$\langle \tilde{w}_i - \tilde{w}_k, h(z) \rangle = 0 \quad \text{for any } z \in \mathbb{S}^{d-1}. \tag{22}$$

However, we have just seen that the linear span of the image of $h$ is $\mathbb{R}^d$, which implies that $\tilde{w}_i = \tilde{w}_k$. Consequently, we may abuse out notation by setting $\tilde{w}_c = \tilde{w}_i$ if $\mathcal{C}(i) = c$, which yields a new form for (20):

$$\langle \tilde{w}_a - \tilde{w}_b, h(z) \rangle = \langle v_a - v_b, z \rangle, \tag{23}$$

for any $a, b \in \mathscr{C}$ and any $z \in \mathbb{S}^{d-1}$.

**Linear transformation from $v_a - v_b$ to $\tilde{w}_a - \tilde{w}_b$:**  We now prove the existence of a linear map $\mathcal{A}$ on $\mathbb{R}^d$ such that $\mathcal{A}(v_a - v_b) = \tilde{w}_a - \tilde{w}_b$ for any $a, b \in \mathscr{C}$. For this, we prove that the following mapping is well-defined:

$$\mathcal{A} : \sum_{a,b \in \mathscr{C}} \lambda_{ab}(v_a - v_b) \mapsto \sum_{a,b \in \mathscr{C}} \lambda_{ab}(\tilde{w}_a - \tilde{w}_b). \tag{24}$$

Since the system $\{v_a - v_b | a, b\}$ is not necessarily linearly independent, we have to prove that the mapping is independent of the choice of the linear combination. More precisely if for some coefficients $\lambda_{ab}, \lambda'_{ab}$

$$\sum_{a,b \in \mathscr{C}} \lambda_{ab}(v_a - v_b) = \sum_{a,b \in \mathscr{C}} \lambda'_{ab}(v_a - v_b) \tag{25}$$

holds, then it should be implied that

$$\sum_{a,b \in \mathscr{C}} \lambda_{ab}(\tilde{w}_a - \tilde{w}_b) = \sum_{a,b \in \mathscr{C}} \lambda'_{ab}(\tilde{w}_a - \tilde{w}_b). \tag{26}$$

Assume that (25) holds. Then, the difference of the two sides is:

$$0 = \sum_{a,b \in \mathscr{C}} (\lambda_{ab} - \lambda'_{ab})(v_a - v_b). \tag{27}$$

Taking the scalar product with an arbitrary $z \in \mathbb{S}^{d-1}$ and using the linearity of the scalar product gives us:

$$0 = \langle \sum_{a,b \in \mathscr{C}} (\lambda_{ab} - \lambda'_{ab})(v_a - v_b), z \rangle = \sum_{a,b \in \mathscr{C}} (\lambda_{ab} - \lambda'_{ab})\langle v_a - v_b, z \rangle. \tag{28}$$

Now using (23) yields:

$$0 = \sum_{a,b \in \mathscr{C}} (\lambda_{ab} - \lambda'_{ab})\langle \tilde{w}_a - \tilde{w}_b, h(z) \rangle = \langle \sum_{a,b \in \mathscr{C}} (\lambda_{ab} - \lambda'_{ab})(\tilde{w}_a - \tilde{w}_b), h(z) \rangle. \tag{29}$$

However, the linear span of the image of $h$ is $\mathbb{R}^d$, which implies that

$$\sum_{a,b \in \mathscr{C}} (\lambda_{ab} - \lambda'_{ab})(\tilde{w}_a - \tilde{w}_b) = 0, \tag{30}$$

equivalent to (26). Therefore, the mapping is well-defined. The linearity of $\mathcal{A}$ follows trivially.

**$h$ is linear:**  Equation (23) becomes:

$$\langle \mathcal{A}(v_a - v_b), h(z) \rangle = \langle v_a - v_b, z \rangle, \tag{31}$$

for any $a, b \in \mathscr{C}$ and any $z \in \mathbb{S}^{d-1}$. Nevertheless, $\{v_a - v_b | a, b \in \mathscr{C}\}$ is a generator system of $\mathbb{R}^d$, and, hence, (31) is equivalent to

$$\langle \mathcal{A}y, h(z) \rangle = \langle y, z \rangle, \quad \text{for any } y \in \mathbb{R}^d \text{ and any } z \in \mathbb{S}^{d-1}. \tag{32}$$

This is further equivalent to

$$\langle y, \mathcal{A}^\top h(z) \rangle = \langle y, z \rangle. \tag{33}$$

Since $y$ is arbitrary, we conclude that $\mathcal{A}^\top h(z) = z$ for any $z \in \mathbb{S}^{d-1}$. Therefore $\mathcal{A}$ is an invertible transformation and $h = (\mathcal{A}^\top)^{-1}$ is linear.

**Proving Thm. 1C case C4:**   We have shown that $\boldsymbol{h}$ is linear. Furthermore, from (31) it follows, by fixing $b$ and defining $\boldsymbol{\psi} = \mathcal{A}\boldsymbol{v}_b - \boldsymbol{w}_b$, that

$$\tilde{\boldsymbol{w}}_a = \mathcal{A}\boldsymbol{v}_a + \boldsymbol{\psi}, \quad \text{for any } a \in \mathscr{C}, \tag{34}$$

which proves case C4 of Thm. 1C.

**Proving Thm. 1C case C2:**   As a special case of the previous one, now we assume that $\boldsymbol{h}(\boldsymbol{z})$ is unit-normalized and maps $\mathbb{S}^{d-1}$ to $\mathbb{S}^{d-1}$. That amounts to $\boldsymbol{h} = (\mathcal{A}^{\top})^{-1}$ being linear, norm-preserving, and therefore orthogonal. Consequently $\mathcal{A}$ is also orthogonal, $\boldsymbol{h} = \mathcal{A}$ and (34) simplifies to $\frac{\beta}{\alpha}\boldsymbol{w}_a = \tilde{\boldsymbol{w}}_a = \mathcal{A}\boldsymbol{v}_a + \boldsymbol{\psi} = \boldsymbol{h}(\boldsymbol{v}_a) + \boldsymbol{\psi}$, which proves C2 of Thm. 1C.

**Proving Thm. 1C case C1:**   We now assume that both $\boldsymbol{h}$ and $\boldsymbol{w}_i$'s are unit-normalized. Consequently, $\boldsymbol{h} = \mathcal{A}$ is orthogonal linear and $\boldsymbol{w}_a = \frac{\alpha}{\beta}\mathcal{A}\boldsymbol{v}_a + \boldsymbol{\psi}$.

Therefore, on one hand, the $\boldsymbol{w}_a$'s lie on a $d$-dimensional hypersphere of radius $\frac{\alpha}{\beta}$ and center $\boldsymbol{\psi}$. On the other hand, by definition, $\boldsymbol{w}_a$'s also lie on the unit hypersphere $\mathbb{S}^{d-1}$.

Since the system $\{\boldsymbol{w}_a | a \in \mathscr{C}\}$ is the bijective affine linear image of the affine generator system $\{\boldsymbol{v}_a | a \in \mathscr{C}\}$, $\{\boldsymbol{w}_a | a \in \mathscr{C}\}$ is also an affine generator system (Lem. 1). Consequently, there could be at most one hypersphere in $\mathbb{R}^d$ which contains all the $\boldsymbol{w}_a$'s. Hence $\frac{\alpha}{\beta} = 1$, $\boldsymbol{\psi} = \boldsymbol{0}$, and $\boldsymbol{w}_a = \boldsymbol{h}(\boldsymbol{v}_a)$, which proves C1 of Thm. 1C.

**Proving Thm. 1C case C3:**   Finally, we assume that $\boldsymbol{w}_i$'s are unit-normalized. As this is a special case of Thm. 1C C4, we know that there exists a constant vector $\boldsymbol{\psi}$ such that:

$$\boldsymbol{w}_a = \frac{\alpha}{\beta}\mathcal{A}\boldsymbol{v}_a + \boldsymbol{\psi}, \tag{35}$$

for any $a \in \mathscr{C}$. We are going to prove that $\mathcal{O} = \frac{\alpha}{\beta}\mathcal{A}$ is orthogonal and $\boldsymbol{\psi} = \boldsymbol{0}$.

Let $\mathcal{O} = \mathcal{U}^{\top}\Sigma\mathcal{V}$ be the singular value decomposition (SVD) of $\mathcal{O}$. Consequently, after premultiplying with $\mathcal{U}$, we receive:

$$\mathcal{U}\boldsymbol{w}_a = \Sigma\mathcal{V}\boldsymbol{v}_a + \mathcal{U}\boldsymbol{\psi}. \tag{36}$$

As orthogonal transformations $\mathcal{U}$ and $\mathcal{V}$ keep their arguments unit-normalized and $\{\mathcal{V}\boldsymbol{v}_a - \mathcal{V}\boldsymbol{v}_b\}$ is still an affine generator system (Lem. 1), we may assume without the loss of generality that

$$\boldsymbol{w}_a = \Sigma\boldsymbol{v}_a + \boldsymbol{\psi}, \tag{37}$$

for any $a \in \mathscr{C}$, where all $\boldsymbol{v}_a$'s and $\boldsymbol{w}_a$'s are unit-normalized.

Let us assume that $\boldsymbol{\psi} \neq \boldsymbol{0}$. In that case both sides of (37) can be scaled such that the offset $\boldsymbol{\psi}$ has unit norm. In this case $\boldsymbol{w}_a$'s are no longer on the unit hypersphere, but they instead have a mutual norm $r$. Assuming that the diagonal elements of $\Sigma$ are $\boldsymbol{\sigma} = (\sigma_1, \ldots, \sigma_d)$, this is equivalent to:

$$r^2 = \|\Sigma\boldsymbol{v}_a + \boldsymbol{\psi}\|^2 = \|\Sigma\boldsymbol{v}_a\|^2 + 2\langle\Sigma\boldsymbol{v}_a, \boldsymbol{\psi}\rangle + \|\boldsymbol{\psi}\|^2 \tag{38}$$

$$= \langle\boldsymbol{v}_a \odot \boldsymbol{v}_a, \boldsymbol{\sigma} \odot \boldsymbol{\sigma}\rangle + \langle\boldsymbol{v}_a, 2\boldsymbol{\sigma} \odot \boldsymbol{\psi}\rangle + 1, \tag{39}$$

where $[\boldsymbol{x} \odot \boldsymbol{y}]_i = x_i y_i$ is the elementwise product. Eq. (39) is equivalent to the following:

$$(\boldsymbol{v}_a \odot \boldsymbol{v}_a)^{\top}(\boldsymbol{\sigma} \odot \boldsymbol{\sigma}) + \boldsymbol{v}_a^{\top}(2\boldsymbol{\sigma} \odot \boldsymbol{\psi}) - r^2 = -1. \tag{40}$$

Collecting the equations for all $a \in \mathscr{C}$ yields:

$$\mathcal{D}\begin{pmatrix} \boldsymbol{\sigma} \odot \boldsymbol{\sigma} \\ 2\boldsymbol{\sigma} \odot \boldsymbol{\psi} \\ r^2 \end{pmatrix} = -\mathbf{1}_{|\mathscr{C}|}, \tag{41}$$

where $\mathcal{D}$ is the following $|\mathscr{C}| \times (2d+1)$ matrix:

$$\mathcal{D} = \begin{pmatrix} \cdots\cdots & \cdots\cdots & \cdots \\ (\boldsymbol{v}_a \odot \boldsymbol{v}_a)^{\top} & \boldsymbol{v}_a^{\top} & -1 \\ \cdots\cdots & \cdots\cdots & \cdots \end{pmatrix}. \tag{42}$$

By Assum. 2, the left $|\mathscr{C}| \times 2d$ submatrix of $\mathcal{D}$ has full rank of $2d$. Consequently, the solution space to the more general, linear equation $\mathcal{D}\boldsymbol{t} = -\mathbf{1}_{|\mathscr{C}|}$, where $\boldsymbol{t} \in \mathbb{R}^d$, has a dimensionality of at most 1.

Using the unit-normality of $v_a$'s, we see that $(v_a \odot v_a)^\top 1_d = 1$. From this, it follows that the solutions are exactly the following:

$$t = \begin{pmatrix} \gamma \cdot 1_d \\ 0_d \\ \gamma + 1 \end{pmatrix}, \quad \text{where } \gamma \in \mathbb{R}. \tag{43}$$

Therefore, for any solution of (41) there exists $\gamma$ such that:

$$\sigma \odot \sigma = \gamma \cdot 1_d \tag{44}$$
$$\sigma \odot \psi = 0_d. \tag{45}$$

However, as the original transformation $\mathcal{A}$ was invertible, all singular values $\sigma_i$ are strictly positive and, thus, it follows that $\psi = 0$. Technically speaking, this is a contradiction to our initial assumption that $\psi \neq 0$. All in all, it follows that $\psi = 0$ is the only possibility.

Therefore, (37) becomes:

$$w_a = \Sigma v_a, \tag{46}$$

where all $v_a$'s and $w_a$'s are unit-normalized. Following the same derivation yields:

$$1 = \|\Sigma v_a\|^2 = (v_a \odot v_a)^\top (\sigma \odot \sigma), \tag{47}$$

or, after collecting the equations for all $a \in \mathscr{C}$:

$$\mathcal{B}(\sigma \odot \sigma) = 1_{|\mathscr{C}|}, \tag{48}$$

where $\mathcal{B}$ is the $|\mathscr{C}| \times d$ matrix

$$\mathcal{B} = \begin{pmatrix} \cdots\cdots\cdots \\ (v_a \odot v_a)^\top \\ \cdots\cdots\cdots \end{pmatrix}. \tag{49}$$

By Assum. 2, $\mathcal{B}$ has full rank, thus, there is at most one solution to the equation $\mathcal{B}t = 1_{|\mathscr{C}|}$. Due to the unit-normality of $v_a$'s, this solution is exactly $t = 1_d$. However, as the singular values $\sigma_i$ are all positive, the only solution to $\sigma \odot \sigma = 1_d$ is $\sigma = 1_d$. This is equivalent to saying that $\mathcal{O} = \frac{\alpha}{\beta} \mathcal{A}$ is orthogonal.

Furthermore, $h = (\mathcal{A}^\top)^{-1} = (\frac{\beta}{\alpha} \mathcal{O}^\top)^{-1} = \frac{\alpha}{\beta} \mathcal{O}$.

$\square$

# B   Additional experimental results

In Tab. 2, we present additional ablation studies exploring the effect of varying the levels of concentration for $v_c$ across the unit hyper-sphere. We do not observe any significant impact on the $R^2$ scores from more concentrated cluster centroids $v_c$.

Table 2: Identifiability in the synthetic setup. Mean $\pm$ standard deviation across 5 random seeds. Settings that match our theoretical assumptions are ✓. We report the $R^2$ score for linear mappings, $\tilde{z} \to z$ and $w_i \to v_c$ for cases with normalized (o) and unnormalized (a) $w_i$. For unnormalized $w_i$, we verify that mappings $\tilde{z} \to z$ are orthogonal by reporting the mean absolute error between their singular values and those of an orthogonal transformation.

| | | | | | | normalized $w_i$ cases | | | | unnormalized $w_i$ | |
| | | | | | | $R_o^2(\uparrow)$ | | $\text{MAE}_o(\downarrow)$ | | $R_a^2(\uparrow)$ | |
| $N$ | $d$ | $|\mathscr{C}|$ | $p(v_c)$ | $p(z|v_c)$ | M. | $\tilde{z} \to z$ | $w_i \to v_c$ | $\tilde{z} \to z$ | $w_i \to v_c$ | $\tilde{z} \to z$ | $w_i \to v_c$ |
|---|---|---|---|---|---|---|---|---|---|---|---|
| $10^3$ | 5 | 100 | Uniform | vMF($\kappa{=}10$) | ✓ | $98.6_{\pm0.01}$ | $99.9_{\pm0.01}$ | $0.01_{\pm0.00}$ | $0.00_{\pm0.00}$ | $99.0_{\pm0.00}$ | $99.9_{\pm0.00}$ |
| $10^3$ | 5 | 100 | Laplace | vMF($\kappa{=}10$) | ✓ | $98.7_{\pm0.00}$ | $99.5_{\pm0.00}$ | $0.01_{\pm0.00}$ | $0.00_{\pm0.00}$ | $99.1_{\pm0.00}$ | $99.8_{\pm0.00}$ |
| $10^3$ | 5 | 100 | Normal | vMF($\kappa{=}10$) | ✓ | $98.2_{\pm0.01}$ | $99.2_{\pm0.01}$ | $0.01_{\pm0.00}$ | $0.00_{\pm0.00}$ | $99.2_{\pm0.00}$ | $99.8_{\pm0.00}$ |

# C  Acronyms

**CL** Contrastive Learning

**DGP** data generating process

**ICA** Independent Component Analysis

**LVM** latent variable model

**SSL** Self-Supervised Learning

**vMF** von Mises-Fisher

