# OpenReview forum: "DIETing: Self-Supervised Learning with Instance Discrimination Learns Identifiable Features"
_NeurIPS.cc/2024/Workshop/UniReps — UniReps_

### Official Review · Reviewer_2FjE · 2024-09-29
**Evaluation of the paper "DIETing: Learning Recognizable Features through Self Supervised Learning with Instance Discrimination"**

**Rating:** 8
**Confidence:** 4

**Review:**

This article introduces DIET, a simplified self supervised learning (SSL) method that focuses on instance recognition and claims to be able to recover recognizable cluster based latent representations. DIET has demonstrated its competitiveness by reducing the complexity of typical SSL pipelines, while also ensuring the recognizability of features. The author provides theoretical recognizability guarantees and empirical verification of synthesized data.

---

### Official Review · Reviewer_TrcK · 2024-10-05
**Interesting work that provides theoretical guarantees for a novel SSL method called DIET**

**Rating:** 9
**Confidence:** 3

**Review:**

This paper provides identifiability theory for a novel yet simple SSL method called DIET.

**Pros:**
* This paper provides theoretical guarantees on the identifiability of features learned using a recently introduced SSL method called DIET.
* The theoretical results are interesting, and are empirically verified under settings that both conform to and violate the theoretical assumptions.
* The assumptions are clearly stated.

I can't comment on the correctness of the theoretical results and proofs.

**Cons:**
* I feel like there are some missing pieces of important information/discussions that would help me better understand the intuitions and the results of this paper. Please see the questions/comments below.

**Further questions/comments:**
* Lines `42 - 46`: The authors stated that the theoretical guarantees presented in this paper apply to SSL representations being used post-training (i.e., without removing the projection head/network), whereas other SSL theoretical/identifiability results apply to the representations with the projection network removed. One of the cited work is Zimmerman et al., 2021, which makes this comparison incorrect based on my understanding of that work. The theoretical guarantees in Zimmerman et al., 2021 also apply to representations being used post-training.
* Why did the authors take a cluster-centric approach for the theory of the DIET method? What's the motivation for taking this approach?
* Assumption 1 (i): why does the set of cluster vectors {$\mathbf{v}_c$} need to be sufficiently large? What happens if there are fewer semantic classes?
* Lines `90 - 91`: "for any $i, j, C(i) = C(j)$ implies $w_i = w_j$". Why is this the case? My understanding is that each $w_i$ corresponds to an instance with instance label $i$ whose ground truth latent $z$ is drawn from a vMF distribution around the cluster vector $v_c$ of class $c=C(i)$. Why do two instances with labels $i, j$ that have the same semantic class $c$ imply that their $w$ is identical?
* Experimental results: why does the identifiability suffer when there's a lower number of clusters (i.e., when Assumption 1 (i) is violated)? While the authors provided justifications for this phenomenon (lines `124-127`), which includes the content-style partitioning of the latents and insufficient augmentation overlap, it is still not very clear to me why the identifiability would depend on the number of semantic classes/clusters. Further explanations on this empirical observation would be helpful.

---

### Official Review · Reviewer_K3DH · 2024-10-07
**Very solid, self-contained theoretical analysis with experimental verification of SSL representation formation, however, in a very simplified setup.**

**Rating:** 7
**Confidence:** 3

**Review:**

## 1. Summary
The work presents a theoretical proof with additional experimental verification of the effectiveness of the self-supervised learning method DIET in a simplified, synthetic data-based setup. Specifically, the work proves the development of cluster-based latent representations under the instance-based SSL method DIET when instance augmentations and intra-class samples are drawn from von Mises-Fisher distributions.

## 2. Strengths and Weaknesses
The work addresses a significant problem related to how and which representations are formed using a type of self-supervised learning method.

The problem statement, the approach taken, the results, as well as the work's limitations, are all stated extraordinarily clearly.

A weakness is the potentially oversimplified problem setup for the theoretical analysis. However, given that it is a submission for the Extended Abstract track, I believe this work to be a promising first step. The authors fully acknowledge the limitations and propose next steps that are fully in line with my own questions regarding the work (reducing constraints and studying the effect of more complex and realistic data generation processes).

## 3. Questions

(More targeted for future work:)
How would you improve on the chosen data generation process to cover more realistic scenarios and are major changes expected?

## 4. Limitations

The work fully acknowledges its own limitations in its limitations section, primarily regarding scale and the limited complexity and introduced constraints for the theoretical study.

My main concern (which is also stated as a limitation in the work itself) is the validity of assuming a von Mises-Fisher distribution as the data-generating process, specifically the identical treatment of instance augmentations and other intra-class samples. This assumption might be an oversimplification and render the results trivial.

## 5. Ethical Concerns

None

## 6. Soundness

4 - Excellent

## 7. Presentation

4 - Excellent

## 8. Contribution

3 - Good

---

### Official Review · Reviewer_cf57 · 2024-10-07

**Rating:** 5
**Confidence:** 2

**Review:**

The paper presents a proof and empirical validation on synthethic data that, under certain constraints, DIET (a self-supervised learning approach with sample index prediction) can recover latent representations and cluster centroids.

Admittedly, I didn't follow all details, but equations 8 and 9 in the appendix suggest that with the given constraints, the linear relationship between learned latent representations and original latent representations holds regardless the choice of f(x) (learned representations). That makes me question why f(x) is learned at all, and what the impact of learning it is. In this regard, I would have expected a more detailed analysis of architectural choices and parameters in the evaluation and in-depth discussion beyond "We also observe that for a higher concentration of samples around $v_c$ (i.e. $\kappa=50$) as well as lower number of clusters (i.e. |C|=10), identifiabilitty suffers..." (I assume $\kappa$ is the amount of samples per class).

The main body of the paper is somewhat hard to follow without the appendix, e.g., section 2 defines cases C3 and C4 in Theorem 1, while C1 and C2 are defined in the appendix (which is referred to before the theorem with "Thm. 1C in Appx. A").

---

### Decision · Program_Chairs · 2024-10-10

**Decision:**

Accept

**Comment:**

In light of the positive reviewers' feedback and relevancy of the submission, we are pleased to accept this paper for presentation at UniReps 2024. We kindly ask the authors to incorporate the reviewers' suggestions and feedback in the final camera-ready version of the manuscript, especially making it more self-contained and accessible.